# Learning risk management of geohazards in practice with free and open-source web-GIS based platform: RISKGIS

Zar Chi Aye<sup>1,2</sup>, Roya Olyazadeh<sup>1,2</sup>, Marc-Henri Derron<sup>1,2</sup>, Michel Jaboyedoff<sup>1,2</sup>, Johann Lüthi<sup>2</sup>

<sup>1</sup>Institute of Earth Sciences, University of Lausanne, Lausanne, Switzerland

<sup>2</sup>Faculty of Geosciences and Environment, University of Lausanne, Lausanne, Switzerland

Correspondence to: Zar Chi Aye (zarchi.aye@unil.ch)

Abstract. "How do environmental risk systems function?" is the main underlying question to be answered by students learning environmental risk. Under the framework of the Innovative Teaching project, an open-source, interactive and collaborative web-GIS application (RISKGIS) is developed for students in learning and understanding of environmental risk

- systems with a focus on geohazards and risk. The aim is for students to better understand and become familiarized with approaches used by experts as well as for teachers to better evaluate and monitor student learning, through a practical application with real case studies and hand-on exercises. To identify the possibility and applicability of the developed learning platform, a series of practical exercises is carried out with undergraduate and postgraduate students during the spring and autumn semesters of the environmental risk, and advanced risk and vulnerability courses of the University of
- Lausanne. A total of three exercises are conducted starting from the rapid risk calculation (individual) exercise to more complex risk management (individual and group) exercise with different case studies and functionalities of the learning platform. Depending on the exercises, students are asked to answer the test quiz, feedback questionnaires and group reports (if requested) through the Moodle platform to evaluate the performance of the students, exercises and the RISKGIS platform. Average feedback results from three different exercises revealed that students found the exercises useful and interesting,
- while a user satisfaction score of 7/10 and a system usability scale (SUS) of 64/100 is achieved, showing that several aspects of the RISKGIS learning platform could be further improved with suggestions and feedback of the students.

#### **1** Introduction

As natural hazards are location dependent, risk management activities can benefit from geographical representations. Geospatial technologies such as Remote Sensing (RS) and Geographic Information Systems (GIS) are increasingly utilized as useful decision support tools in natural hazards and risk management (Coppock, 1995; Thomas and Kemec, 2007; Manfré et al., 2012; van Westen, 2013). The rapidly growing technology such as GIS plays an important role in understanding and managing natural hazards, referenced to a geographical location (Carrarra and Fausto, 1995; Peggion et al., 2008). For example, in the process of risk assessment, GIS can be used to calculate potential damages in affected areas caused by a hazard event, helping risk managers and decision makers to take appropriate preventive measures. With regards to teaching

5

and learning within natural hazards, environment and risk, GIS applications have been widely used as a supporting platform for effectively achieving goals of the science education such as utilization of the power of technology, development of data analysis and (spatial) thinking skills (Barstow, 1994; Gutierrez et al., 2002; Bodzin and Anastasio, 2006; Mitchell et al., 2008). Over the years, GIS technology benefited from the evolution of web, and with the advancement of technologies, it has become possible not only to visualize and disseminate spatial data and information over the web but also to analyse, model and process interactively (Dragićević, 2004). Subsequently, Spatial Data Infrastructure (SDI), web-GIS and geo-visualisation

tools have been applied in numerous works of the natural hazards community (Sugumaran et al., 2000; Mansourian et al.,

2006; Müller et al., 2006; Sutanta et al., 2009; Aye et al., 2016a).
Particularly in teaching and learning, web-GIS platforms have become increasingly applied in instructional settings to overcome barriers such as cost, accessibility to hardware and software, and lack of technical support (Audet and Paris, 1997;

- Carver et al., 2004; Bodzin and Anastasio, 2006). These platforms can be easily accessible from web browsers without needing to use limited resources of the Lab and purchase GIS software. Moreover, the progress in web-GIS helped advancing collaborative and participative approaches in risk management (Dragićević and Balram, 2004). This in turn
  facilitated the centralised sharing and analysis of data and information, especially in performing collaborative group
- exercises with students under the integrated risk management framework. Due to these several advantages, web-GIS platforms designed for different purposes in teaching and learning have been applied for subjects related to climate change, natural hazards and environmental risk management (Tobita et al., 2008; Frigerio and van Westen, 2010; Joost et al., 2012; Pechanec and Vávra, 2013; Careaga, 2014). Learning risk management through interactive web-GIS tools can be regarded as
- a technology-supported, active learning activity (Aye et al., 2016c; Kos, 2009). Bonwell and Eison (1991, p. 2) defined "active learning" as an activity in which students are involved "in doing things and thinking about the things they are doing". This activity is meaningful and important, which contributes to the understanding of concepts to be learned (Wiggins and McTighe, 1998). Besides, the application of open-source technology combined with e-learning tool such as Moodle (Modular Object-Oriented Dynamic Learning Environment) could bring innovative pedagogical advantages and values to
- conventional curricula and classroom practice, such as interactive and active learning, collaborative interaction between students, accessibility to a wide range of online resources from discussion forums to teaching materials, assessment and feedback (Brandl, 2005). These kinds of interactive learning platforms and activities could help increasing the motivation and interest of students in their studies.
- Even though some teaching projects using web-GIS exist (Painho et al., 2001; Bodzin and Anastasio, 2006; Wang et al., 2016), they are almost exclusively used for the visualization and dissemination of spatial data with little capacity for edition and simulation of data. In particular for the estimation and management of natural hazards, risk analysis tools are generally intended for experts, rarely in the form of web-GIS, and inadequate for teaching (OFEV, 2016; RoadRisk, 2017; ZHA, 2017). Developing such a platform from scratch would be a considerable work, and therefore, taking the advantage of open-

source and previous related works done by authors (Aye et al., 2016a; Aye et al., 2016b; Olyazadeh et al., 2016), a platform named "RISKGIS" is developed for teaching under the framework of the Innovative Teaching project (FIP) of the University of Lausanne. RISKGIS is an open-source, collaborative web-GIS based learning platform, and is adapted for students in learning and understanding of environmental risk systems with a focus on geohazards and risk. This platform is particularly

- aimed to apply in some exercises of the "environmental risk" and "advanced quantitative risk and vulnerability" courses of the Bachelor and Master of Sciences programs in the Faculty of Geosciences and Environment. The exercises of the courses are usually carried out individually by students in a paper-based manner, for example, an exercise for risk calculation of the affected buildings in a given area exposed to an event of debris flows. Beyond this conventional approach, with the aid of the RISKGIS's specific modules, exercises are taken place throughout the semesters on a web-GIS platform. This allows
- students to visualize, interrogate and edit spatial data not only for risk calculation individually but also for working collaboratively in groups to propose and evaluate different risk reduction solutions using cost-benefit and multi-criteria evaluation methods integrated in the learning platform. Following in this paper, we present the context and framework of the teaching project in Sect. 2. Section 3 further presents the background framework and methods of the RISKGIS learning platform. In Sect. 4, different scenarios carried out with students using RISKGIS are described along with a discussion of
- collected feedback results from students. Finally, we conclude the paper with reflection on the achieved outcomes and potential perspectives of the presented learning platform.

#### 2 Context and framework of the project

The learning platform for students is proposed and designed, based on open-source modules and web-GIS applications developed within previous research works of authors (Aye et al., 2016a; Aye et al., 2016b; Olyazadeh et al., 2016) and the experiences of testing a prototype collaborative platform with master students during a course on risk communication at the University of Lausanne in April 2015 (Aye et al., 2016c). In this exercise, students played the roles of different risk management stakeholders, and feedback was collected from them on different aspects of the exercise and platform. One of the aspects is whether students would like to do other exercises with such kind of interactive tools, and favourable feedback were achieved for students with experience in GIS. This serves as one of the needs why the RISKGIS platform is introduced

- to teaching and learning in risk management of geohazards, by adapting existing approaches to the ones used in Switzerland, which are relevant to environmental risk (120-160 students) and advanced quantitative risk and vulnerability (10-15 students) courses of the University of Lausanne. The environmental risk course sets the goal to introduce environmental risks and their implications on the society from a quantitative viewpoint, while the advanced quantitative risk and vulnerability course is dedicated to the in-depth understanding of risk notions for a thorough risk calculation of natural hazards and it
- 30 presents methods of Swiss Confederation among others including expert approaches such as impact-probability matrices.

A large number of students follow the environmental risk course, and therefore, it is difficult to carry out a personalized follow-up of the exercises and to evaluate the level of understanding of the students. Besides, in risk analysis, solutions are often not unique and have to be discussed during the corrections, and as students, feedback on the proposed solutions is expected of teachers. Since the analysis of environmental risk is very much related to spatiality, it is difficult to deal with realistic, generally large and complex, case during a course. Hence, until now, the solution was to deal with synthetic cases,

- which are sometimes too simplistic. In addition, the time spent to perform calculation by hand leaves little room for analysis of the results, which is furthermore less interesting with synthetic cases. Therefore, with this new web-GIS platform, real events of natural disasters can be used as case studies, greatly facilitating the management of spatial data and numerical calculation part. Students can also change different parameters of the calculation process to create various risk scenarios, and
- this provides more room for the analysis and understanding of different outcomes. Moreover, with specific functionalities and modules of the learning platform, students can learn an overview of the risk management framework, starting from the risk analysis to the decision making process, like experts in real world with hand-on exercises.

#### 2.1 Envisaged pedagogical scenarios

The aim is to create progressive learning, starting with risk calculation and ending with risk management. Some steps can be carried out "by hand", however, others require the use of the web-GIS platform for centralized sharing of data and results, covering a larger geographical area. The use of this new technique permits to go further in the understanding and the assimilation of the issues, the proposition and the discussion of the solutions. Currently, students use GIS in several courses. GIS software is installed locally in computer rooms. These programs allow the display of geographic data and offer the possibility to carry out numerous processing (mappings, queries, calculations, etc.). Nevertheless, they are sometimes

- complex to use and they do not allow a common (centralized) sharing of solutions. In parallel to traditional GIS, web-GIS tools have been developed, but only simple operations are usually allowed for most of the times (visualization of data and simple queries) via a web browser. The web portals of the cantons (<u>www.geo.vd.ch</u>) and confederation (<u>www.map.geo.admin.ch</u>) belong to this category, and some have been developed to facilitate education (EBIBALPIN, 2015; Tectonics, 2015). Therefore, the development of a new interactive web-GIS platform (RISKGIS) for teaching would
- enable students not only to easily visualize data but also to perform operations (computation, introduction and edition of spatial data, simulations, etc.), and to save data in a central database. Students can connect to the platform with an individual account and see different interfaces/functionalities depending on their roles and exercises. The platform does not require any local installation or license, and can be freely accessible through a simple web-browser.
- Three possible scenarios (exercises) are defined, built on the workflow of risk management framework (Fig. 1). The first scenario is to learn how to use the web-GIS platform with a simple risk calculation on an individual basis. The second scenario is to develop a risk reduction strategy by carrying out risk calculation (before and after consideration of measures), to carry out a simple cost-benefit calculation of the proposed strategy, and to confront it with others. This is a group work

exercise and each group is asked to submit a group report with an explanation and justification of their choices and results. The third scenario focuses on the importance of different actors, agendas and interests in group decision-making. It adds an additional component, Multi-Criteria Analysis (MCA), for selection of different risk reduction strategies by considering preferences of involved actors on different criteria. In this scenario, students play roles of different actors (such as geologists,

- 5 planners, environmental associations, mayor and population) in groups, and final decision is taken in a participative manner through discussion in the classroom. All scenarios use real events of natural disasters as case studies. The scenarios are then complemented with test quizzes and different (online) questionnaires to evaluate the learnt aspects on risk management, exercise and platform for improvements. The learning workflow of the RISKGIS platform including methods used is presented later in Sect. 3, and further detailed on the structure, content and results of scenarios (exercises) carried out with
- 10 students are then described and discussed in Sect. 4.