# Peer review of "Learning risk management of geohazards in practice with free and open-source web-GIS based platform: RISKGIS"

_Natural Hazards and Earth System Sciences, 2017_

## Referee Comment (RC1) · Anonymous Referee #1 · 6 Apr 2017

This paper presents an open-source web GIS platform designed for conducting risk assessment and cost-benefit analysis of mitigation measures. It is an additional paper in a series of papers already published with a similar content (see e.g. Aye et al., 2016c). The tool follows the method, which has become standard in Switzerland for prioritizing mitigation method by the Federal Office for the Environment. As such the presented method is not new. RISKGIS appears to have an appealing design. The project seems to have a lot of potential to make courses on geohazard risk more interesting and hands-on. This supports the generally effective "learning by doing" approach, while better preparing students for work after university. As such the work is very valuable in the education of future natural hazard specialists.

However, I doubt the scientific contribution of this paper, which is one of main goals of NHESS. Furthermore, the scientific quality is poor, since this paper only describes the tool, its application in case studies and the response of students regarding the performance of the software. The conclusions of students are similar to conclusions already published in Aye et al., 2016c, which reads as "could be further improved". Therefore, the novelty of this paper could be questioned. Although the paper is well structured and concepts and exercises are described in detail so that the reader can get a good idea of the tool and the students' work sequence, I cannot recommend the publication unless substantial scientific findings are included in the paper. As our comments indicate, the used terminology should be critically checked since it is not used consistently throughout the paper.

Additionally, please have a native speaker do a detailed revision of the language. Text flow and comprehension need to be improved

Please also note the supplement to this comment:
http://www.nat-hazards-earth-syst-sci-discuss.net/nhess-2017-85/nhess-2017-85-RC1-supplement.pdf

[Figure]

**Supplement:**

**Review: Learning risk management of geohazards in practice with free and open-source web-GIS based platform: RISKGIS**

In particular, we have the following comments:

| page | line | comment |
| --- | --- | --- |
| 1 | 1 | "…with **a** free and…" |
| 1 | 9 | "…is developed for students **studying** environmental risk…" |
| 1 | 10 | "…become **familiar** with…" |
| 1 | 12 | "…hand**s**-on…" |
| 1 | 12 | "…To identify the **potential** and **practicality** of the …" |
| 1 | 14 | "… semesters of the *Environmental Risk* and *Advanced Risk and Vulnerability* courses **at** the University…" |
| 1 | 15-16 | "…are conducted starting **with** the rapid risk … exercise **and moving on to the** more complex risk … exercise **incorporating** different case studies…" |
| 1 | 17 | "…are asked to **take a test**, **complete** feedback questionnaires **or write** group reports **on** the Moodle platform **in order** to evaluate the **exercises, the RISKGIS platform and the performance of the students**…" |
| 1 | 20 | "…of 64/100 **are** achieved…" |
| 1 | 21 | "…and feedback **from** the students…" |
| 1 | 26 | "...van Westen, 2013). **Rapidly developing technologies** such as GIS **play** an…" |
| 2 | 2 | "…achieving goals **of science** education such as **utilisation of technology and** development of…" |
| 2 | 4 | "…the evolution of **the** web and with the advancement of **technology** it has…" |
| 2 | 10 | Either "in teaching and learning" or "in instructional settings" – you are saying the same thing twice in one sentence. |
| 2 | 11 | Accessibility to hardware: you still need a computer to access a web-GIS, so how is the need for hardware reduced? |
| 2 | 12-13 | "…platforms can be easily accessed from…" or "…platforms are easily accessible from web browsers without **purchasing** GIS software. …" |
| 2 | 12 | "limited resources of the Lab": What resources, what lab?  See suggestion above (p. 2, lines 12-13). |
| 2 | 19-20 | "interactive, active, activity" in the same sentence – try to eliminate at least one. E.g. "task" instead of activity. |
| 3 | 14 | ..and in the following paper: Consider using "exercise" instead of "scenario" since scenario has a different meaning in the risk context. |
| 6 | 20 | Explain what you mean with "Jigsaw". |
| 7 | 5-6 | eliminate parentheses: either use the word or don't (exception: IRM). |
| 8 | Figure 2 and rows 3-5 | Choose more unmistakable terminology for the figure (and in general). E.g. instead of "alternative formulation" use "planning of measures" or "choosing a course of action". Also, make sure your explanation of the figure (rows 3-5) uses the same words as appear in the diagram. Add numbers to the figure so it is very clear which step is which and the reader doesn't have to do the matching herself. |
| 8 | 10 | What is meant by hazard layer? I would rather suggest the term "scenario". |
| 9 | 8 | The value 5000000 CHF needs to be further explained. I recommend |

| | | |
|---|---|---|
| | | to replace it with a factor since 5000000 CHF is a value used in Switzerland. The VSL approach reflects the societal willingness to pay for averting a fatality, which is closely related to a specific country. |
| 9 | 11 | 12 hours in each of 365 days per year. |
| 9 | 15-21 | The unit for individual risk should be 1/year in my eyes. It translates as how often per year, a particular person (a person living in object $i$) is likely to die. With the unit "deaths/year" you are suggesting that the individual risk says how many people die per year, which is what the collective risk (non-monetised) is about. By using the unit 1/year, you can eliminate the irritating "1" in equation 3. |
| 9 | 28-30 | year in citation UNISDR is missing |
| 10-11 | 26-29 | Use a variable (e.g. CBR) instead of an actual ratio ($N/K_{tot}$) to name the cost-benefit ratio in equation 5.
 $K(j)$ is confusing, especially since up until now, $j$ has been the index for a particular hazard. Use $K_{tot}$ throughout.
 Even if it is fairly obvious, explain R(before) and R(after) when you list your variables below equation 5. |
| 11 | 14 and following | Alternative and criteria are not capitalised in this context. |
| 11 | 23 | Idea**l** not idea |
| 11 | 20 | Explain $L_p(x_p^*)$ in equation 7. |
| 12 | 10-13 | Why are some words italic? Also the case in following lines. |
| 12 | 16 | Why did so few students comply? Could it be that only a certain "type" of student took the quiz and answered the questionnaire, thus distorting your results? |
| 12 | 23 | What is the benefit of a remote area? |
| 12 | 26 | Wouldn't that be a return period of 500 years, then? 25 years may not make a big difference on that scale but it is confusing for the reader. |
| 12 | 27 | The term "location site effects" does not make sense. |
| 14 | 3 | Figure 4: There should not be any red bars (even if they are very slight) for a 0 value. Use the word points (or an equivalent) instead of notes (in the text above and below the figure as well as in the figure itself). |
| 14 | 6 | As it turns out in line 11 and following, you are not asking questions but giving statements which students can agree or disagree with. Hence, please change the word "question" as well as the abbreviation "Q" for the statements (applies to the rest of the paper, too).
 Mention the Likert scale here as well, not only later on in the paper. It may give the impression, that you are using two different approaches. |
| 14 | 9 | You might mention that a SUS score above 68 is considered above average… |
| 15 | 9-16 | Consider putting these pros and cons into a bullet point list for a better overview. Consider it, too, for the presentation of the other results in the following paper. |
| 20 | 19 | Was the consideration of ecological aspects already part of this exercise? |
| 26 | 12 | This question does not fit well with the scale "not at all to absolutely". Consider rephrasing it or adding a second scale. |
| 26 | 20 | What is moderate, what is high severity? |
| 27 | 11 | Explain your scale of 1 to 5 in words in the text, not just the figure caption. Again, not all the questions are suitable for "not at all to absolutely". |

| 27 | Fig.15 | Try to avoid 3D graphs. |
|---|---|---|
| 28 | 8 | "…cost was the better choice despite the limitation of risk for human beings." I don't understand this argument. |
| 28 | 23 | Either use the word collaborative or don't but don't put it in parentheses – this makes for cumbersome reading. |
| 28-29 | | A detailed repetition of the questionnaire results is not appropriate in the discussion. Generalise and mention only the key points. Also, the suggestion of the exercises running parallel to the course should not appear for the first time in the discussion – this belongs with the questionnaire results and can be briefly mentioned here as a potential future development for the course.. |
| generally | | <li>Especially for key vocabulary, choose one term and stick with it throughout. E.g. protection measures – why confuse the reader with "alternatives"? Or "scenarios" for "exercises" – simply use exercises in the whole paper to keep things clear.</li><li>Give a brief description of the Innovative Teaching Project.</li><li>What is the purpose of the questionnaires concerning the group work (e.g. figure 15)? What do you learn from the answers and how is this information useful?</li><li>In the case of the Brienz case study: Was the real solution presented and discussed in class? That would nicely round off the exercise, showing students what was decided by the experts and explaining why.</li><li>Could the tool be adapted for real decision makers in natural hazard risk? Perhaps that could be a future aim for the project.</li> |

---

## Referee Comment (RC2) · Anonymous Referee #2 · 6 Apr 2017

Dear authors the manuscript 'Learning risk management of geohazards in practice with free and open-source web-GIS based platform: RISKGIS' describes and provides an overview of an teaching project with students at bachelor and master level. The web-GIS based platform aims that students learn and understand environmental systems with a focus on geohazards and risk as well as to get familiar with different expert approaches applied in Switzerland. The platform should also allow the lecturer to evaluate the performance of the students and the web-tool. The platform is tested and illustrated by three different examples applied in courses of the University of Lausanne presenting different settings such as bachelor and master level, individual task for students or group work, and high and low number of students. The main results of this study are the evaluation by the students of the different courses using the platform.

The idea of the web-GIS platform in teaching and preparing students for their applied work after university has a high potential and is attractive for students by using the approach 'learning by doing'. However, the presentation of this study in the manuscript is a detailed description of the teaching projects, yet it lacks of new scientific insights in the field of education, web-GIS based platforms or risk analysis considering also the existing publication of this project (Aye et al 2016c). For example section 2 highlights some pedagogical concepts and aims, however the questions of the courses evaluation answered by students is not clearly related to these concepts and seems to be strongly based on the standard lecture/exercise evaluation at universities. I miss a clear and structured approach to evaluated a) pedagogical concept/aim of the project and b) the usability of the web-platform. I cannot recommend the current version of the manuscript for publication due to the missing added value for the scientific community as described above. Furthermore, I would recommend re-structuring the manuscript according to focused research questions, considering a clear structure for methods, results and discussion (e.g. questions of the evaluation are part of the methods section) and re-writing a critical discussion section according to the new research questions and highlighting the pro and cons of the web-GIS platform in education. Moreover, I suggest language editing and check used technical terms according the consistency within the manuscript. Some further comments are highlighted in the attached file.

Please also note the supplement to this comment:
http://www.nat-hazards-earth-syst-sci-discuss.net/nhess-2017-85/nhess-2017-85-RC2-supplement.pdf
* * *
[Figure]

**Supplement:**

[Figure]

[revised manuscript text omitted]
 ([www.geo.vd.ch](www.geo.vd.ch)) and confederation ([www.map.geo.admin.ch](www.map.geo.admin.ch)) belong to this category, and some have been developed to facilitate education (EBIBALPIN, 2015; Tectonics, 2015). Therefore, the development of a new interactive web-GIS platform (RISKGIS) for teaching would enable students not only to easily visualize data but also to perform operations (computation, introduction and edition of spatial data, simulations, etc.), and to save data in a central database. Students can connect to the platform with an individual account and see different interfaces/functionalities depending on their roles and exercises. The platform does not require any local installation or license, and can be freely accessible through a simple web-browser.

Three possible scenarios (exercises) are defined, built on the workflow of risk management framework (Fig. 1). The first scenario is to learn how to use the web-GIS platform with a simple risk calculation on an individual basis. The second scenario is to develop a risk reduction strategy by carrying out risk calculation (before and after consideration of measures), to carry out a simple cost-benefit calculation of the proposed strategy, and to confront it with others. This is a group work

[Figure]

[Figure]

exercise and each group is asked to submit a group report with an explanation and justification of their choices and results. The third scenario focuses on the importance of different actors, agendas and interests in group decision-making. It adds an additional component, Multi-Criteria Analysis (MCA), for selection of different risk reduction strategies by considering preferences of involved actors on different criteria. In this scenario, students play roles of different actors (such as geologists, planners, environmental associations, mayor and population) in groups, and final decision is taken in a participative manner through discussion in the classroom. All scenarios use real events of natural disasters as case studies. The scenarios are then complemented with test quizzes and different (online) questionnaires to evaluate the learnt aspects on risk management, exercise and platform for improvements. The learning workflow of the RISKGIS platform including methods used is presented later in Sect. 3, and further detailed on the structure, content and results of scenarios (exercises) carried out with students are then described and discussed in Sect. 4.

[Figure]

Figure 1: Illustration of three scenarios based on different stages of risk management.

**2.2 Pedagogical added values and improved learning experience**

RISKGIS brings the pedagogical added values such as the possibility to propose, discuss and compare different solutions of a problem during the courses as well as providing teachers with a better feedback on the students' level of understanding,

considering that the classroom is equipped with a reduced number of assistants at the institute. Besides, with RISKGIS, it has become possible to do exercises with more illustrative examples (such as real cases with no longer synthetic cases), putting students in an active mode which leads them out of a certain passivity of directed synthetic exercises. In addition, the confrontation of different scenarios and the discussion of the issues result in an in-depth learning of the concepts dealt within
the concerned courses in a much more effective way than at present.

Four important aspects can be highlighted in improving the learning experience of students through the use of RISKGIS: working on a concrete situation (the real case study), analysing it (calculation on the platform and debriefing in groups), generalizing of other possible situations (collective discussion with the teacher), and then experimenting these conclusions
with new situations (several case studies are developed). This reflects the four moments of experiential learning, according to Kolb (1984), which allows to develop an in-depth learning of the concepts and skills analysed. Specifically, the following learning strategies are considered to develop experiential learning for students:

- The **actual case study** permits to illustrate the activities carried out by students, by giving them a little more sense and allowing them to make comparisons with reality.

- The **discussions** between students and the teacher make it possible to confront different points of view. Indeed, students can be able to decentre their own answers and benefit from the ideas of their colleagues and the teacher. The discussions also promote students to involve more actively, and thus, support their learning.

- The different **reflective moments** encourage students to have a look back on their work and to analyse their experiences in depth during the various activities.

- The **Jigsaw** allows students to develop expertise concerning the importance of different actors, agendas and interests in decision making. This activity demands the involvement of the students since they are responsible for capturing a role (theme) and then realizing it in different groups. In addition, feedback given from and between peers can strengthen and support their understanding.

- The **visualization of risk situations** using web-GIS throughout the world (such as real situations in Switzerland
and the Alps for debris flows, the Haiti earthquake or the analysis of destructions due to typhoon Haiyan) allows to better understand risk analysis by anchoring to real cases.

**3 Background framework and methods**

The structure of the RISKGIS learning environment (Fig. 2) is based on two main components of risk management framework: risk analysis (estimation) and risk reduction (treatment). Risk estimation is defined as "the process of deriving a
measure of the probability and severity of loss to the elements at risk by the integration of hazard and consequence analysis" (Crozier and Glade, 2005), and this can be performed quantitatively or qualitatively, depending on the study scale, data availability and aims of the analysis (Aye et al., 2016a). In this paper and developed learning framework, we apply risk

[Figure]

concepts and calculation approaches used in Switzerland (Bründl et al., 2009; EconoMe, 2015; Valdorisk, 2015). The outcomes of this process serve as an important input for decision making in risk management, depending on whether risks are tolerable or intolerable. Risk treatment is a process for risk modification and involves the selection and implementation of one or more options (Crozier and Glade, 2005). For students, within the RISKGIS learning environment, the focus is placed on preparedness (and prevention) phase of Integrated Risk Management (IRM) cycle before an event is occurred. There are many options (alternatives) which can be applied in order to reduce the risk based on factors contributing to risk, for example, modifying the impacts of affected elements by reducing their vulnerability or reducing the intensity of the hazard through the implementation of some structural control measures in the area. This IRM approach requires a combination of measures for risk reduction (Holub and Hubl, 2008), and therefore, there is a need for coordinated efforts between responsible authorities and organizations in selection of efficient and effective measures (De Marchi and Scolobig, 2012; Aye et al., 2016b). Consequently, this learning platform is designed in a collaborative manner so that students can reflect and learn this important issue of risk management in reality. Besides, through role playing, students can think and reflect content in a real world context beyond the classroom setting. For example, according to a mayor or a responsible actor's prioritization on different decision criteria, what is the most favourable alternative solution in selection of risk reduction strategies?

[Figure]

[Figure]

[Figure]

**Figure 2: Workflow of the RISKGIS learning environment and its components.**

The workflow of the learning platform is organized in six steps as follows: 1) risk estimation; 2) formulation of alternatives; 3) cost estimation of alternatives; 4) recalculation of risk after alternatives; 5) cost-benefit calculation of alternatives; and 6) selection of alternatives with Multi-Criteria Analysis.

**Step 1: Risk estimation**

In this step, risk is estimated by using «Risk Calculator» tool of the RISKGIS learning platform, where default parameters (i.e. hazard, building and vulnerability parameters) can be changed to create different risk scenarios and analyse the variation in risk. In the learning platform, the ***risk of the buildings*** $R_{i,j}$ (CHF/year) is calculated for each affected building $i$ of the whole building layer and for a hazard layer $j$ as follows (modified from EconoMe, 2015; Valdorisk, 2015):

$$R_{i,j} = f_j \times PrA_j \times V_{i,j} \times W_i, \tag{1}$$

[Figure]

[Figure]

where $f_j$ is the frequency (inverse of the return period $T$) of the hazard $j$; $PrA_j$ is the probability of spatial occurrence of the hazard $j$; $V_{i,j}$ is the respective vulnerability value of the building $i$ depending on the intensity class of the hazard $j$ in which it is exposed; and $W_i$ is the monetary value of the building $i$.

For the monetarized ***collective risk of persons*** (inside the buildings), $R_{i,j}$ (CHF/year) is calculated for each affected building $i$ of the whole building layer and for a hazard layer $j$ as follows (modified from Bründl et al., 2009 and EconoMe, 2015; Valdorisk, 2015):

$$R_{i,j} = f_j \times PrA_j \times Ppr_i \times \lambda_{i,j} \times N_i \times 5000000 \text{ CHF,} \qquad (2)$$

where $f_j$ is the frequency (inverse of the return period $T$) of the hazard $j$; $PrA_j$ is the probability of spatial occurrence of the 10 hazard j; $Ppr_i$ is the probability of presence of persons in the building object $i$ (for example, 0.5 for a person who is present in the building for 12 hours per day); $\lambda_{i,j}$ is the lethality of persons in the building object $i$ for the considered hazard $j$; $N_i$ is the number of persons in the building object $i$; and 5000000 CHF is the marginal cost that the society is willing to pay to avoid the death.

For the ***individual risk*** (death/year), it is calculated the same as Eq. (2), except the number of persons and the marginal cost of a person are not included (modified from Bründl et al., 2009 and EconoMe, 2015; Valdorisk, 2015):

$$R_{i,j} = f_j \times PrA_j \times Ppr_i \times \lambda_{i,j} \times 1, \qquad (3)$$

where $f_j$ is the frequency (inverse of the return period $T$) of the hazard $j$; $PrA_j$ is the probability of spatial occurrence of the hazard j; $Ppr_i$ is the probability of presence of the person in the building object $i$ (for example, 0.5 for a person who is 20 present in the building for 12 hours per day); $\lambda_{i,j}$ is the lethality of the person in the building object $i$ for the considered hazard $j$; and 1 is a person.

**Step 2: Formulation of alternatives (design of preliminary measures)**

After identifying areas at risk, the next step is to formulate possible alternatives based on the outcomes of the previous step. A combination of risk reduction measures is possible within an "alternative" (both structural and/or non-structural 25 measures). Potential risks can be reduced based on the contributing factors such as hazard, exposure of elements-at-risk and vulnerability through the implementation of effective risk management strategies. Structural measures are defined as "any physical construction to reduce or avoid possible impacts of hazards, or application of engineering techniques to achieve hazard-resistance and resilience in structures or systems" (UNISDR, p. 28). Non-structural measures are "any measures not involving physical construction that uses knowledge, practice or agreement to reduce risks and impacts, in particular through 30 policies and laws, public awareness raising, training and education" (UNISDR, p. 28). Here, students can design/propose

alternatives through the web-GIS mapping interface using available sketching tools and measures in the learning platform. Examples of measures are dike, tunnel, stabilization works, protection nets against rock falls and debris flows, reforestation, maintenance of the protection forests and so on.

**Step 3: Cost estimation of alternatives**

After identifying possible risk reduction measures in the affected area, the potential cost of measures is estimated for their proposed alternatives. The annual cost of the measure is calculated as follows (Bründl et al., 2011):

$$K_{tot} = K_b + K_u + K_r + \frac{(I_0 - L_n)}{n} + \frac{(I_0 + L_n)}{2} \times \frac{p}{100}, \qquad (4)$$

where $K_b$ is the operating cost; $K_u$ is the maintenance cost; $K_r$ is the repairs cost; $I_0$ is the initial investment cost; $L_n$ is the residual value after *n* years (in general, $L_n = 0$); and *p* is the interest rate fixed at 2%. Within the learning platform, measures and associated default parameter values for cost estimation are obtained from EconoMe (OFEV, 2016) and Valdorisk (Valdorisk, 2015). Depending on the "mode" of the cost estimation process (auto or manual), these default parameters can be changed accordingly by users of the RISKGIS platform.

**Step 4: Risk re-calculation after alternatives**

The next step is to re-calculate risk (in a preliminary manner) after the consideration of an alternative to know how much risk has been reduced in the affected area. In order to re-calculate risk, students can create new layers of hazard and/or building, depending on their designed alternative. For example, if an alternative includes measures which modify (or reduce) the intensity level and spatial location (polygon shape) of hazard, then a new hazard layer is created and modified accordingly based on the original hazard layer. Or if an alternative considers measures which affect the location of buildings (for example, relocation of houses), a new building layer can be created to remove relocated houses from the original building layer. If an alternative includes measures which change the vulnerability, then vulnerability parameters can be directly modified during the re-calculation of risk.

**Step 5: Cost-Benefit calculation**

As a final step of the selection process of alternatives, a cost-benefit report is prepared to compare the risk before and after the consideration of an alternative. For this purpose, the cost-benefit ratio is calculated in the learning platform as follows (Bründl et al., 2011):

$$\frac{N}{K_{tot}} = \frac{R(before) - R(after)}{K(j)} = \frac{R(v)}{K(j)}, \qquad (5)$$

where $\frac{N}{K_{tot}}$ is the cost-benefit ratio of the measure or the combination of measures, and it is profitable if this ratio is $\geq 1$; $R(v)$ is the annual reduction of risk before and after the implementation of measures; and $K(j)$ is the annual cost of the planned alternative $j$ (protection measures).

**Step 6: Selection of alternatives with Multi-Criteria Analysis**

Going a step further in the decision making process and taking the advantage of the collaborative and centralized web-based environment, the RISKGIS learning platform integrates MCA approach for evaluation of alternatives based on various decision criteria. The decision making process can benefit from using MCA methods. These methods consider different alternatives with the aim of addressing trade-offs between them by including additional criteria (such as social and environmental) than the traditional cost-benefit criteria (Munda, 2004). Besides, these approaches allow to represent conflicting views of involved actors and facilitate the decision making process (Kiker et al., 2005). In the learning platform, Compromise Programming (CP) method (Zeleny, 1973; Simonovic, 2010) is applied for ranking of alternatives. By means of distance, this method identifies the closest alternatives to the ideal solution, and this ideal solution is a vector of best values of evaluated decision criteria resulting from a payoff (or performance evaluation) matrix. This payoff matrix $A$ is an evaluation matrix of $m$ Alternatives against $n$ Criteria, and is represented as follows in Eq. (6):

$$
\quad A = \left[a_{ij}\right] = \begin{bmatrix} a_{11} & a_{12} & ... & a_{1n} \\ a_{21} & a_{22} & ... & a_{2n} \\ ... & ... & ... & ... \\ a_{m1} & a_{m2} & ... & a_{mn} \end{bmatrix}, \tag{6}
$$

where $a_{ij}$ is the evaluation value of Alternative $i$ for Criteria $j$; $m$ is the number of alternatives and $n$ is the number of criteria. The distance measure of an alternative $L_p(x)$ is "a function of the criteria values themselves, the relative importance of the various criteria to the decision makers ($\alpha_i$), and the importance of the maximal deviation from the ideal solution ($p$)" as shown in following Eq. (7) (Simonovic, 2010, p. 274):

$$
\quad L_p\left(x_p^*\right) = Min\left\{ L_p(x) = \left[ \sum_{i=1}^{r} \alpha_i^p \left( \frac{A_i^* - A_i(x)}{A_i^* - A_i^{**}} \right)^p \right]^{\frac{1}{p}} \right\}, \tag{7}
$$

where $A_i(x)$ is the evaluation value of an alternative $x$ for the considered criterion $i$; $A^*$ is the maximum function value of the considered criterion; $A^{**}$ is the minimum function value of the considered criterion; $\alpha_i$ is the weight (relative importance) of the considered criterion; $p$ is the importance of the maximal deviation from the idea solution and $r$ is the

number of criteria. The alternative with the minimum distance value to the ideal situation is considered as the "best compromise solution".

**4 Scenarios with students using RISKGIS learning platform**

**4.1 Scenario 1: Getting started with RISKGIS**

The purpose of this scenario is to introduce the RISKGIS platform to students in learning and understanding of risk estimation process with a case study. This exercise was carried out with Bachelor students of the environmental risk course during the exercise session on 14 March 2016. The exercise is composed of two stages: exploration of available maps and tools, and risk estimation. Necessary data and maps are prepared and uploaded to the learning platform by teachers and assistants beforehand. Before starting the exercise, the short introduction of the platform and the task was presented to students. The *tutorial* documentation and a step-by-step *video* demonstration are also made available. Then, by following the given step-by-step instructions of the tutorial and data of the study area, each student logs in to the platform and performs a simple risk calculation. At the end of the exercise, in the Moodle (www.moodle.org), students are asked to answer a *quiz* to check whether they understood the purposes of different functionalities in the platform as well as a *feedback questionnaire* to obtain the user evaluation feedback on the platform. The access to the platform, test quiz and feedback questionnaire were opened for two weeks starting from the day of the exercise. Out of the registered 82 students, 30 students answered the quiz and 23 students responded the feedback questionnaire. The results are presented below in Sect. 4.1.3.

**4.1.1 Study area**

An example study area of the exercise is in Jomsom town located in the mountainous area of Nepal at an altitude of about 2700 m in Mustang District, which is a famous touristic area known for hikers in Himalayas. Figure 3 shows an overview of areas around Jomsom in Mustang District, where landslides affected some agricultural fields located in the valley (left) and one of the schools in the area (right). Jomsom was slightly affected by the 2015 earthquake (7.8 Magnitude, VIII of Modified Mercalli Intensity Scale) and many buildings were damaged, however, no injuries were reported in the town (Nelson News, 2015; SMH, 2015). We chose it as a study area not only because it is a remote area affected by earthquake but also because of the availability of OpenStreetMap data. According to the Global Seismic Hazard Map (Giardini et al., 2003), the seismic hazard of this area is estimated to be severe showing Peak Ground Acceleration (pga) of 0.368 g (i.e. VIII Scale of MMI) with 10% probability of exceedance in 50 years, corresponding to a return period of 475 years. In this exercise, we imagine that if this area may be affected by location site effects (such as amplifications of shaking) targeting the centre of Jomsom, then what would be the risk for buildings and people? For data inputs, buildings are extracted from OpenStreetMap (www.openstreetmap.org) and the earthquake intensity map is designed based on the geological characteristics and DEM

(Digital Elevation Model) of the area by an expert. The default vulnerability values for damages and death rates are derived

[Figure]

[Figure]

from ATC-13 report (Rojahn and Sharpe, 1985). Here, we assume that buildings in Jomsom are unreinforced masonry houses, and the casualty rates are tied to the damage states of the houses.

[Figure]

[Figure]

**Figure 3: Photos of the areas around Jomsom in Mustang District (Nepal), showing landslides which affected some agricultural fields located in the valley (left) and one of the schools in the area (right).**

**4.1.2 Stages of the scenario**

In this first stage of the exercise, students are asked to start with adding layers of building and earthquake intensity maps to the web-GIS map interface of the learning platform. For this purpose, building and intensity maps are uploaded into the platform beforehand. The task of this stage is for students to identify and visualize the potential Jomsom areas affected by the earthquake by simply overlaying these two layers and visualizing the buildings which are being touched by different levels of earthquake intensity. Besides, to learn basic components of GIS, students are also asked to explore available tools in the platform such as adding and removing layers to and from the map panel, zoom and feature information tools, location search box and so on. In the second stage, students estimate risk of buildings and humans for the whole study area by using the «Risk Calculator» tool of the RISKGIS learning platform (see Fig. 8 for an example illustration, however, only "Risk Analysis" menu is visible and enabled in this exercise). Here, students can change the default building information such as price per square meter (USD), number and exposition of people (hours) as well as vulnerability information (i.e. degree of damages and casualties in percentage in relation to a certain level of intensity and damage state) to analyse the variation in risk depending on the input parameters. For this risk calculation, a return period of 1 year is used in this exercise. The risk results could therefore be divided by the corresponding return period (for example, 475 years) to obtain risk of a certain return period.

**4.1.3 Results and discussion**

For the test *quiz* in Moodle, 10 questions were asked to students concerning different functions and interfaces of the platform, mainly for the risk calculation module such as definition of risk, the intensity level which has the highest risk per year, number of buildings exposed to the earthquake, and so on. Out of 30 students' answers, the global average score was 9.39 out of 10 notes (maximum). This means most of the students learned and understood correctly of how the platform

[Figure]

[Figure]

functions. The following Fig. 4 shows the total number of students respective to the interval of notes (1 note is given for each question and a total of 10 questions were asked).

[Figure]

**Figure 4: Number of students with respective notes (1 note is given for each question and a total of 10 questions were asked).**

The *user evaluation feedback* questionnaire is composed of different sections: 1) purpose of the platform (in a few words); 2) a simple 5-point, ten-item question of SUS (System Usability Scale) to obtain a global view of the usability assessment (Brooke, 1996); 3) an overall satisfaction score (on a scale of 1-10) of using the platform; and 4) aspects of the platform to be improved and supplementary suggestions/comments. Out of 23 responses (Fig. 5), we achieved an average SUS score of 68.41/100 and the overall satisfaction of the platform was 7.13/10. The ten items of the SUS (Q1-Q10) are listed as follows, representing positive and negative items:

Q1.    I think that I would like to use this system frequently.

Q2.    I found the system unnecessarily complex.

Q3.    I thought the system was easy to use.

Q4.    I think that I would need the support of a technical person to be able to use this system.

Q5.    I found the various functions in this system were well integrated.

Q6.    I thought there was too much inconsistency in this system.

Q7.    I would imagine that most people would learn to use this system very quickly.

Q8.    I found the system very cumbersome to use.

Q9.    I felt very confident using the system.

Q10.  I needed to learn a lot of things before I could get going with this system.

[Figure]

[Figure]

Figure 5: Feedback on the platform (overall) based on a 5-point, ten-item scale of SUS. The answers were rated by a scale of 1 to 5 (Strongly disagree to strongly agree).

To sum up the feedback of students on written sections, students mentioned the purpose of the platform is to evaluate and model risk (materials and humans) of a study zone easily and rapidly in function of pre-defined parameters and existing data, helping to better understand and analyse risks, and to find more effective risk reduction solutions based on outcomes. It was also mentioned that, thanks to the computer power, it is possible to simply modify numerous variables and consider multiple scenarios of risk, by making the process simplified and more accessible to the quantification and representation of different types of risks. Regarding the aspects to be improved, some students found the platform very interesting and easy to use, and that it is difficult to make a criticism because the platform looks very much like other software that they are already using, while some mentioned that usage of the platform on portable systems such as tablets and smart phones can be improved and that the attribute columns of the risk layer should be made clearer. It was also mentioned that being able to adjust the size of the window panels (i.e. Data view, legend and layers) would be helpful. For additional comments and suggestions, students stated to make a much longer and deeper exercise with the application as well as to create a good structure of saving output layers in a more structured way, while they have also showed their interests of using it for other possible exercises with study areas in the Alps to better get an idea of the terrain. According to the responses, the purpose of the platform and exercise was well-understood by students, and favourable feedback were received on using the RISKGIS learning platform in doing this very first exercise with students.

**4.2 Scenario 2: Selection of a risk reduction strategy using RISKGIS**

The learning objective of this scenario is for students to develop a better understanding in risk management using the RISKGIS learning platform with a real case study (for example, debris flows event in Brienz, Switzerland). This second

exercise was again performed with same Bachelor students of the environmental risk course during the course and exercise sessions on 18 April 2016. The structure of the exercise follows the workflow of the learning framework illustrated in Fig. 2 without the last MCA component. This exercise was carried out in small groups (no more than 5 students in each and students chose their group partners via Moodle), and a login account was created for each group so that results can be shared and visualized within the same group. By following the given *tutorial* exercise and data of the study area, each group of students proposes its own alternatives (risk reduction measures) by carrying out risk calculation (before and after consideration of alternatives) and performs cost-benefit analysis of the proposed alternatives. To complete the exercise, students are required to submit via the Moodle platform: a *group report* on the obtained results, an *exercise* and a *user evaluation feedback* individually to evaluate the performance of the exercise and platform. The access to the platform, submission of the group reports and feedback questionnaires were opened for three weeks. Out of the registered 85 students, 69 students participated and formed 15 groups. Before the deadline of the exercise, 14 groups submitted their reports, 55 and 52 students responded the exercise and user evaluation feedback questionnaires respectively. The results of students' feedback are presented and discussed in Sect. 4.2.3.

**4.2.1 Study area**

In this exercise, we study the village of Brienz located on the Shore of Brienz Lake in the canton of Bern, Switzerland (Fig. 6a). In 22-23 August 2005, after a long lasting rainfall (more than 300 mm in 3 days), landslides and debris flows occurred and caused extensive damage to this village (Fig. 6b). Approximately 70,000 m$^3$ of debris was transported into a densely populated part of the village (PLANAT, 2016). Two people died, about 30 houses were damaged completely or partly, and 300 people were needed to be evacuated in a very short time. The total damage was over 30 million Swiss francs (Hitz and

Hahlen, 2014), and this event was one of the many disastrous events which occurred during August 2005 in the Bernese Oberland and the Swiss Alps. After this event, a flood prevention project was initiated and the designation of the protection works was inaugurated in August 2013, which involved costs of about 35 million Swiss francs (Haberle, 2014). In this exercise, we reconsider that this August 2005 event has just happened due to the prolonged rainfall in the area. Then, students try to answer questions such as risks of buildings and people in this area, measures to reduce risks and their cost effectiveness and so on. For the study data, buildings are extracted from OpenStreetMap and the debris flow intensity map (with a return period of 100 years) is derived based on the expert knowledge and hazard assessment carried out after August 2005 (Hitz and Hahlen, 2014). The default vulnerability values for damages and causalities are obtained from EconoMe (OFEV, 2016). Here, we consider that buildings in this study area fall into the category of individual houses, exclusively for the residential use (such as villas and chalets).

[Figure]

[Figure]

[Figure]

**Figure 6: (a) Brienz with the two landslides (red) in the catchment areas of the Trachtbach and Glyssibach torrents (Source: Haeberle, 2014). (b) Debris flow event and its damages along Glyssibach, Brienz in August 2005 (Source: PLANAT, 2016).**

**4.2.2 Stages of the scenario**

In this exercise (Fig. 7), each group of students is asked to submit a group report on the considered study area by answering the following questions with an explanation and justification of their choices and results:

- risks of buildings and people,
- needed measures to be placed in order to reduce risk,
- estimated cost of the proposed alternative,
- risks after the implementation of the proposed alternative, and
- profitability of the proposed alternative in terms of cost and benefit.

[Figure]

[Figure]

**Figure 7: Structure of the exercise illustrating different steps of the exercise, in which students (in groups) use the RISKGIS and Moodle platforms for performing tasks and activities.**

First of all, students get themselves familiarized with the study area by exploring available data (including building and
hazard intensity maps), identifying and visualizing the potential areas affected by debris flows. Then, students estimate risk
by selecting necessary parameters of the «Risk Calculator» tool as shown in Fig. 8 to answer the first question of the
exercise. In the next step 3, based on the outcomes of the risk estimation process, we ask each group to design and propose
their own alternative to reduce risk in the area. Here, we define "alternative" as a combination of risk reduction measures,
which could be both structural and/or non-structural measures. Students can design their alternatives based on the available
measures in the learning platform and answer the question of why their proposed alternatives should be considered and
implemented. After formulating measures, in the step 4, students calculate annual cost of their proposed alternatives by
entering estimated values of different parameters as shown in Eq. (4). Then in the step 5, students re-calculate risk after
consideration of their proposed alternatives by creating and modifying new hazard and/or building layers (Fig. 9), depending
on the measures they considered for risk reduction. Finally in the last step, students generate a cost-benefit report by
selecting risk scenarios they calculated before and after the consideration of proposed alternatives, and answer the question
of whether their proposed alternative is profitable based on the obtained cost-benefit ratio.

[Figure]

[Figure]

[Figure]

**Figure 8: Illustration of the popup interface of the «Risk Calculator» tool for creation of a new risk scenario in the RISKGIS platform. This tool is accessible through "Risk Analysis" menu located on the top toolbar of the center map panel. "Input Data" menu contains tools to access input layers (such as hazard and building maps), while "Risk Reduction" menu is composed of tools to create new alternatives, calculate cost of alternatives, and generate cost-benefit report.**

[Figure]

**Figure 9: Illustration of the popup interface of the "Editing existing feature" tool for modification of a layer's feature (for example, in editing the hazard intensity map) in the RISKGIS platform.**

[Figure]

[Figure]

**4.2.3 Results and discussion**

In the *exercise feedback* questionnaire, we asked students: 1) to explain what they have learnt from doing this exercise (in few words); 2) to respond to five questions on a simple Likert scale of 1 to 5 (Strongly disagree to Strongly agree) for the feedback on the exercise; and 3) to mention aspects of the exercise to be improved and if there are any supplementary suggestions/comments. Fig. 10 shows the average ratings of the five following questions (Q1-Q5) asked:

Q1.   This exercise is interesting.

Q2.   This exercise is useful for my learning and understanding.

Q3.   This exercise is helpful in understanding of how real situation works.

Q4.   This exercise stimulates my interests in risk management.

Q5.   I want to do practical exercises with such interactive tools.

Out of the obtained 55 responses, 70% of students agreed that the exercise was interesting and useful for their learning and understanding. About 60% of students mentioned that this exercise helped in understanding of how the reality works, while 55% and 53% of students indicated that this exercise stimulated their interests in risk management and that they want to do practical exercises using this kind of interactive tools. To sum up their feedback on what they have learnt in this exercise, students mentioned that this exercise allowed them to familiarise concretely with risk management by engaging them in the elaboration of a solution for a real case, in which they have learnt to identify potential risks of a given area, develop an effective and realistic alternative, and finally to estimate the profitability of their solution. They also stated that they learned to consider different possible alternatives which are financially and physically adequate to reduce risk, as well as advantages and disadvantages of measures taken (such as financial and ecological aspects), while providing a better understanding of the notions of risk calculation and cost-benefit analysis for the exam. Besides, students indicated that they realized the complexity of risk systems in practical cases (such as difficulty in quantifying the impact of a natural disaster and finding a perfect alternative), and that they have seen a direct application of the course, making them possible to put into practice and understand theories seen in course during the semester in a very concrete way, thanks to the automation of the entire computational part, which is not always easy to assimilate theories without concreate case studies.

[Figure]

**Average Score (55 responses)**

[Figure]

Figure 10: Exercise feedback on a 5-point Likert scale (Strongly disagree to strongly agree).

Regarding aspects of the exercise to be improved, students mentioned mainly to provide them with additional sources and reference values for a better cost estimation of measures (for example, price per cubic meter of a dike). They also highlighted that it would have been useful to present more concretely on how various measures reduce the hazard or risk of a given area (for example, effectiveness of a dike in comparison with a concrete dam or the maintenance of a forest in an area prone to debris flows). In addition, students suggested to make smaller groups (no more than 3) and distribute the exercise on several sessions (for example, 4 x 1 hour) with a possibility to select a case study among a predefined list with different subjects and countries so that they can learn more about measures and their effectiveness in different cases. Besides, they have also stated that the use of the RISKGIS platform should be presented during the course (than the video tutorial) to better understand how to use it and interpret the results. For additional comments, students mentioned it was a very good idea to use an application like this as it allowed to integrate theories in practice, and that it would be more interesting to do more exercises in this genre and less exercises on the paper. While they appreciated for being "put into real situation", they indicated that more data and information should be provided to estimate important parameters for calculating risk or costs, in particular cost of measures and their effectiveness, which was the main issue they encountered in this exercise. It would have been helpful to mention, during the course, a list of the various protective measures with some information (advantages, disadvantages, typical use, costs, lifetime, etc.).

Regarding the *user evaluation feedback*, we used the same SUS questionnaire (Q1-Q10 as in Sect. 4.1.3 of the first scenario)

to evaluate the performance of the platform. For a total of 52 responses, we achieved an average SUS score of 58.03/100 and the overall satisfaction of the platform was 6.12/10, which is lower than the results of the first scenario. This is mainly because of several issues we discussed above in the exercise feedback (as pointed out by students). To summarize students'

feedback on different sections, students mentioned that the main objective of the platform is the apprehension and management of environmental risks in the context of natural hazards, and thanks to it, several hypotheses can be formulated to find interesting solutions. Though it is very easy to use, they need prior knowledge to obtain plausible results, and that it would be interesting to make this tool accessible to users unfamiliar with geo-social concepts of risk, in particular by integrating a workflow or a section explaining different functions that RISKGIS can implement. For the technical aspects, students suggested that it would be good to be able to style measures (for example, dike in blue and tunnel in black) for visual appearance and to create shapes other than polygons in designing measures. It was also mentioned that the interface could be more intuitive with addition of keyboard shortcuts, better presentation of system messages, grouping of overlay layers and ability to save the entire work as a project with layers added so that it would not necessary to add layers again one-by-one into the map. In addition to these technical aspects, many suggested that estimated cost of measures should be proposed automatically, and that the reduction of areas at risk should be guided (or simulated based on proposed measures). Since it was difficult for them to have an idea of the costs and the effectiveness of the measures envisaged, it would be nice to add a database with this type of information. Some students commented additionally that it is a good idea of the real work done with a lot of potential, and that they will also appreciate for having a user guide to test it in a setting other than the course, which would allow them to know how to create a layer or import data.

**4.3 Scenario 3: Learning the role of different actors in risk management using RISKGIS**

The aim of this scenario is to better understand the complexity and conflicting interests of involved actors in the decision making process of risk management through role-playing of different actors such as technicians, authorities and decision makers. This exercise was carried out with 13 Master students of the advanced quantitative risk and vulnerability course on

14 November 2016, using the RISKGIS platform and a real case study in Brienz, Switzerland. The structure of the exercise is based on the complete learning framework (see Fig. 2), and it was performed individually and in groups (3-4 students per group), depending on different stages of the exercise (Fig. 11). For the individual part of the exercise, students can register themselves through the platform to do the exercise individually, while group logins are given to each group of students for the other parts of the exercise in groups. As in previous scenarios, by following the given *tutorial* exercise, *role description*

sheet and data of the study area, students analyse risk, propose risk reduction measures, perform cost-benefit and multi-criteria analyses to have an overview of risk management framework with involvement of different actors. At the end of the exercise, students are asked to submit an *exercise* and a *user evaluation feedback* to evaluate the performance of the exercise and platform, as well as *group functioning questionnaires* for the group decision making part of the exercise. The results of feedback and group functioning questionnaires are presented and discussed in Sect. 4.3.2.

[Figure]

[Figure]

[Figure]

**Figure 11: Workflow (structure) of the scenario illustrating the different stages of the exercise with involvement of different groups (Source: modified from Aye et al., 2016c). The same case study Brienz (Switzerland) of the second scenario (Sect. 4.2.1) is used.**

**4.3.1 Stages of the scenario**

The same study area and input data of the previous scenario 2 is used (i.e. buildings and debris flow hazard map of Brienz, Switzerland) in this exercise (see Sect. 4.2.1). This exercise includes stages of the previous scenario 2 (Sect. 4.2.2), however, with an additional component of MCA for the group decision making part of the exercise. In the first stage, each student calculates risk individually using default parameters. During the second stage, the class is divided into three groups: geologists, spatial planners and environmental associations. Depending on their played roles and outcomes of the calculated risk in the first stage, each group is asked to: 1) propose an alternative; 2) estimate costs of the alternative; 3) re-calculate risk after the consideration of alternative; and 4) calculate cost-benefit ratio of that proposed alternative. Then in the third stage of the exercise, alternatives proposed by different groups are evaluated and ranked with all involved actors using the Compromise Programming method implemented in the RISKGIS platform. The decision criteria are pre-defined by the moderator (i.e. teacher) beforehand to evaluate the performance of alternatives (which were proposed by each group from the previous second stage). Through the interactive discussion with all groups of students in the classroom, we evaluate and assign performance values of alternatives for each criterion, for example, the cost of the alternative proposed by a certain group of actors is very high. These performance values can be either qualitative or quantitative depending on the type of

[Figure]

[Figure]

criteria. For the qualitative criteria, they vary ranging from 1 (Extremely Low) to 9 (Perfect). For this third stage of the exercise, we divided the class into four groups of actors (i.e. public representatives, mayor and municipal council, geologists and planners, and environmental associations). After that, each group ranks alternatives by assigning weights to criteria (Fig. 12). In other words, depending on their played roles, each group decides and classifies the importance of the criteria (with a scale of 1: the *least important* to 5: the *most important* criteria). These weights are normalized (i.e. weight values are divided by the total) for ranking of alternatives. The ranking of alternatives for each group is then calculated dynamically with the given weight set and evaluation matrix using Eq. (6) and (7). Finally, all groups come together and negotiate to achieve a final weight set for the final decision through a classroom discussion moderated by the student group, who plays the role of mayor and municipal council.

[Figure]

**Figure 12: Illustration of the window panel for assigning weights to criteria (i.e. in the first tab) by a group of actors (with a scale of 1: the least important to 5: the most important criteria) in the RISKGIS platform, which is accessible through "Risk Reduction → Decision Alaysis → Prioritization" menu. The evaluation matrix (Alternatives Vs Criteria) is located in the second tab, and results (ranking of alternatives) can be visualized in the third tab based on assigned weights.**

**4.3.2 Results and discussion**

In this exercise, two types of feedback were collected: 1) exercise and user evaluation feedback for evaluation aspects related to student exercise and user experience of the learning platform, and 2) group functioning feedback for understanding of aspects related to group functioning and group decision making.

[Figure]

[Figure]

**Evaluation feedback**

For the *exercise feedback,* we used the same feedback form of the scenario 2 (see Questions 1-5 of the exercise feedback in Sect. 4.2.3). Out of the 13 responses of students (Fig. 13), the best scores were achieved for Q1 and Q3. About 92% and 85% of students agreed that the exercise was interesting and helpful in understanding of the real world situation. While 62% of students found the exercise useful for their learning and understanding, as well as stimulating their interests in risk management. Finally, 54% of students indicated that they want to do practical exercises with this kind of interactive tools. Regarding what they learned from doing this exercise, it was mentioned that physical issues play only a tiny part while financial aspect plays a very important role with a lot of points to be taken into account, and that a decision process is long and demands for negotiations. It was also stated that they have learnt in reflecting which risk protection measures to take, and to have a general vision with the viewpoints of many involved actors. For the aspects to be improved, students expressed again to have more information on the costs of measures and utilization of the platform.

[Figure]

**Figure 13: Exercise feedback on a 5-point Likert scale (Strongly disagree to strongly agree).**

Regarding the *user evaluation feedback* (see Sect. 4.2.3 for the questions Q1-Q10), we achieved an average SUS score of 64.62/100 and the overall satisfaction of the platform was 6.77/10 for 13 students, which is lower than the results of the first scenario but a bit higher than the second scenario. Nevertheless, good scores were achieved for Q7 (i.e. they imagine that most people would learn to use this system very quickly) and Q8 (i.e. they do not found the system very cumbersome to use). Concerning the purpose of the platform, students mentioned that it is to understand different stages and roles of actors upon creation of a risk reduction project in an area. It was also mentioned that the platform is to perform a quick and simple qualitative analysis of risk with some pseudo-quantitative data, cost and cost-benefit calculation with the ability to compare several alternatives and opinions. Regarding aspects of the platform to be improved, it was stated that certain technical

aspects can be improved (especially in editing features and displaying layers), and that it could be great to add some basic impacts of protection structures (i.e. their costs and effectiveness – how to modify/reduce hazard zones).

**Group functioning feedback**

In this scenario, two group functioning questionnaires (based on a scale of 1 to 5: Not at all to absolutely) are also collected to understand how a group with same and mixed roles of actors functions. The following Fig. 14 shows average scores of the questions (Q1-Q5) asked in this first *questionnaire with same roles*, i.e. for the first round of selection of alternatives (in stage 3 of the exercise) with groups of same actors. The questions are as follows:

Q1.   Does your group struggle with the problem of setting goal (i.e. figuring out the desired outcome)?

Q2.   Do the people in your group all agree upon a common goal?

Q3.   a) Were there conflicts in your group in prioritizing decision criteria?

b) If there were conflicts, what is the level of severity?

Q4.   Do you agree with assigned weights (of your group) on decision criteria?

Q5.   Overall, are you satisfied with functioning of your group?

[Figure]

Figure 14: Group functioning feedback (same roles) based on a 5-point Likert scale (Not at all to absolutely).

Out of the obtained responses, at a moderate-to-high level (scale of 3-to-5), 83% of students mentioned that they had struggles in figuring out the desired outcome. However, about 92% of students indicated that they have come to an agreement upon a common goal. Even though 58% of students had conflicts in prioritizing decision criteria within their groups (with a moderate severity of 38% and a high severity of 0%), 92% of students agreed with assigned weights of their groups on decision criteria, and none of the students were not satisfied with functioning of their groups. In this questionnaire, we also asked students to explain their impressions in doing this exercise with their groups, and students commented that it

was an interesting exercise simulating a real case for students by taking a certain view point of actors in risk management and they had to consider other aspects which they do not necessarily consider. Regarding their opinions of whether group work helps tackling problem at hand, most students gave positive feedback mentioning that the group work can help them to see the aspects which they probably would not see alone and that there has an advantage of thinking in certain aspects because of different opinions. While it was also mentioned that it is difficult to set goals for management as it takes a lot of aspects to consider as well as viewpoints of different groups of actors.

For the second *questionnaire with mixed roles* of actors, Fig. 15 shows average scores (in groups) of the questions (Q1-Q5) asked for the final round of selection of alternatives (in stage 3 of the exercise) with groups of mixed actors. The questions are as follows:

Q1. What is your level of influence in your group?

Q2. Do you agree with the final decision taken?

Q3. Were your opinions/suggestions taken into account?

Q4. Did you feel participated in the decision-making process?

Q5. Overall, are you satisfied with functioning of your group?

[Figure]

**Figure 15: Group functioning feedback (mixed roles) based on a 5-point Likert scale (Not at all to absolutely).**

As can be seen in Fig. 15, the group of community leaders had the lowest level of influence compared to the rest of the groups. Except the technician group of geologists and planners, all groups agreed with the final decision taken (by the mayor with discussion in the classroom). The similar result was achieved for the question of whether their opinions were taken into

account, in which the technician group obtained the lowest score of 2.5. Regarding the participation question, the group of community leaders mentioned that they did not feel participated in the process (i.e. a score of 1.33), following by the technician group with a score of 2.75. Overall, for this final round of selection process, an average score of 4.08 was achieved for the satisfaction with group functioning. In general, amongst other groups, we can see that mayor achieved the highest scores, followed by the group of environmental association. This result shows the consistency between collected feedback and exercise, since the final decision taken by the mayor is the solution proposed by the group of environmental association. To sum up the additional questions we asked in this questionnaire, students mentioned that mayor took the final decision by discussing with all groups, and for mayor, cost was the better choice despite the limitation of risk for human beings. It was also stated that group decision making could be improved by having more information of the considered problem, dealing more with all aspects (which they could not due to time limitations) and defining the common goal amongst involved actors (i.e. for example, saving the lives).

**5 Discussion and conclusion**

In this paper, we presented an open-source web-GIS based platform (RISKGIS) for learning risk management of geohazards. Based on previous related research works of authors (Aye et al., 2016c; Olyazadeh et al., 2016), this platform was adapted to methods and approaches applied in Switzerland for undergraduate and postgraduate students learning in environmental risk, and advanced risk and vulnerability courses at the University of Lausanne. It replaces some exercises of the courses, in particular for studies on large areas which cannot be easily calculated by hand, for the rapid evaluation of risk before and after the consideration of protection measures and to test the efficiency of measures using a simple cost-benefit analysis tool. Furthermore, it integrates a multi-criteria evaluation tool for selection of measures collaboratively in the context of group decision making. This research work was carried out under the framework of the Innovative Teaching project, and aimed at enhancing teaching concepts of risk management through real case studies and the practical application of an innovative web-GIS based platform, without needing to deal with commercial software licences and synthetic case studies for exercises. In this research, taking the advantage of new (collaborative) web-GIS capabilities, students can define endangered zones spatially and calculate cost of a disaster under different scenarios considering protective measures in groups. The classical

GIS systems do not allow such collaborative activities, which is indispensable to risk management of natural hazards, between students through exchanges of risk management proposals or between students and the teaching team. Besides, learning of students can be evaluated and monitored via the web-GIS platform.

Three scenarios (exercises) were developed and applied for student learning using the RISKGIS platform with real case studies. For the first exercise, a rapid risk calculation tool of the platform was introduced to Bachelor students of the environmental risk course, and favourable responses were achieved from students mentioning their interests in performing more in-depth exercises using the platform. Students also agreed that most would learn to use this platform very quickly and

it was not very cumbersome to use, while the support of a technical person might be needed for some, since students were not given training or use of the platform before doing the actual exercise (except only a short presentation of the exercise and platform). Afterwards, a much longer second exercise was carried out with same students (in groups) starting from the risk calculation to the selection of protective measures, in which students are required to submit a group report with justifications of their choices and results. Overall, feedback analysis of this exercise showed that 70% of students found the exercise interesting and useful, while about 53% to 60% of students mentioned that it reflected the real situation, stimulating their interests in risk management and doing exercises with such interactive tools. The main issue students encountered in this exercise was the lack of information on costs of protective measures, their effectiveness and modification of hazard zones, as students are still in a learning phase and have limited knowledge unlike experts. This aspect could be significantly improved by providing a more complete data with additional materials and information during or before the exercises. Besides, as one student interestingly suggested, this exercise could be carried out in parallel with the course since the beginning of the semester until the end, so that it would be very interesting and would greatly stimulate students' interests in risk management more, since the theory is not always obvious to assimilate despite a series of exercises during the semester. Finally, the last exercise was performed with master students from advanced quantitative risk and vulnerability course, by following the same structure of the second exercise with additional component on collaborative group decision making using multi-criteria analysis. In this exercise, students (in groups) play roles of different actors in formulation and selection of risk management alternatives. Feedback responses indicated that most students found this exercise interesting and helpful in understanding of the real world situation, while other aspects could be further improved such as information on costs of measures and utilization of the platform. It was also observed that students were satisfied with functioning of their groups (same roles)

despite there were some conflicts in in prioritizing decision criteria and setting upon a common goal in the beginning. However, when different groups come together for final decision, highest satisfaction scores in group functioning were achieved for the mayor's group (i.e. the final decision maker) followed by the group of environmental protection associations (i.e. the winning group out of all proposed alternatives). This was the same for the level of participation in group decision making, and technical group (i.e. geologists and planners) mentioned their disagreement with final decision taken. It was indeed interesting to see how the situation reflects conflicting interests of different actors through the role-playing exercise and interactive discussion in the classroom.

To conclude, feedback obtained from students could be used to further improve the platform and presentation of the exercises, which would increase students' interests, applicability and usability of the presented learning platform. Although there are some aspects to be improved from technical and learning perspectives, students mentioned that these exercises stimulate reflection and creativity by putting them in the position of decision makers facing a disaster. That made them realized that decisions can be complicated and complex not only at the level of what is to be done but also at the level of budget and how one will do to minimize the risk in question. Importantly, it made them possible to have connections with what they learned during the course, permitting also to better analyse and visualize areas at risk for in-depth risk

management linked to a real event. From the teaching perspective, this research work brings several benefits to both students and teachers in terms of feedback by allowing: 1) to confront and discuss risk reduction solutions proposed by students, 2) to compare them with real cases, and 3) to simulate and evaluate the importance of different actors in decision making. This approach attempts to encourage exchange between students or between teachers and students through personal schoolwork and such a collaborative approach is the basic of any modern risk management involving consultation and participative approaches. Moreover, this makes possible to support in-depth learning through the involvement of students in different activities, encouraging their motivation through active involvement and different moments of discussion. Notably, it also creates connections between theory and practice, allowing students to better understand what they have learnt during the course lectures. Last but not least, group activities underpin the development of collaborative skills and decision making which they can later transfer to similar future situations. Regarding the collected feedback of students, some of the improvements are being considered and next exercises using the RISKGIS platform are planned in this spring semester 2017 for Bachelor students of the environmental risk course.

**Author contribution**

All authors have contributed either conceptually or physically to design and materialize this research work, project and article. Zar Chi Aye developed the RISKGIS platform, carried out overall exercises, analysed feedback of students and wrote the manuscript with the kind support and guidance of remaining authors. Roya Olyazadeh assisted in preparing data, tutorial exercises, video demos, SUS and feedback questionnaires via Moodle. Particularly, Roya Olyazadeh designed the quiz test and the first exercise along with tutorial video, based on her previous research work. Selection of study areas, preparation of hazard maps and exercises with students are supported and guided by Michel Jaboyedoff and Marc-Henri Derron. Johann

Lüthi (Pedagogical engineer of the Faculty of Geosciences and Environment) provided suggestions and feedback throughout the project period as well as assistance in particular to aspects related to the Moodle platform.

**Acknowledgements**

The authors would like to thank all participated students of the Environmental Risk and Advanced Quantitative Risk and Vulnerability courses (spring and autumn semesters, 2016) of the University of Lausanne, who performed their exercises using RISKGIS, for giving their valuable feedback and suggestions on the exercises and platform. We would also like to express our thanks to the participated assistants of our Risk Analysis group for translating tutorial exercises into French and assisting students during the exercise sessions of the courses. We acknowledge the funding provided by Fonds d'innovation pédagogique (FIP, 2016) of the University of Lausanne (http://www.unil.ch/fip/), which made possible to carry out this exciting research work for active learning with students in risk management, as well as the support of Deborah Dominguez

(Pedagogical counsellor of CSE, Centre de Soutien à l'Enseignement). Last but not least, we also thank to our former colleague, Pierrick Nicolet, for his kind support and contribution in the proposal submission of this research project.

---

## Author Comment (AC1) · 4 May 2017

**General/Specific comments:**

- *This paper presents an open-source web GIS platform designed for conducting risk assessment and cost-benefit analysis of mitigation measures. It is an additional paper in a series of papers already published with a similar content (see e.g. Aye et al., 2016c). The tool follows the method, which has become standard in Switzerland for prioritizing mitigation method by the Federal Office for the Environment. As such the presented method is not new. RISKGIS appears to have an appealing design. The project seems to have a lot of potential to make courses on geohazard risk more interesting and hands-on. This supports the generally effective "learning by doing" approach, while better preparing students for work after university. As such the work is very valuable in the education of future natural hazard specialists.*

Thanks for your comment on the potential of our project. Regarding the referenced paper *Aye et al., 2016c,* we would like to clarify that this paper is not merely an additional paper with similar contents which we already have published before. RISKGIS was carried out in a different project, and it was rather oriented to teaching and learning in risk management of geohazards. It was based on our previous experience in testing the prototype of *Aye et al., 2016c* with students (which was designed for real experts and decision makers). The results of *Aye et al., 2016c* served as one of the motivations, and RISKGIS is developed using open-source technologies by taking the advantage of our previous research works.

For clarification, new features of RISKGIS are listed below, compared to the prototype of *Aye et al., 2016c*. In this study, RISKGIS is specifically designed for:

- certain courses and practical exercises in university for education;
- risk concepts and methods applied in Switzerland;
- rapid risk estimation based on qualitative (vector) hazard intensity map and OpenStreetMap data;
- cost estimation of measures;
- manually edition of new input maps for risk estimation (such as hazard intensity and buildings map);
- cost-benefit calculation of different risk mitigation scenarios and
- additional features such as registration, customized data sharing and interfaces for students and teachers.

Besides, pedagogical scenarios for progressive learning are designed and implemented in this study, starting with the rapid risk calculation and moving on to the more complex risk management concepts incorporating real events of natural disasters as case studies. In addition, test quiz, group assignments and various questionnaires are integrated via the Moodle platform for the purpose of evaluation.

- *However, I doubt the scientific contribution of this paper, which is one of main goals of NHESS. Furthermore, the scientific quality is poor, since this paper only describes the tool, its application in case studies and the response of students regarding the performance of the software. The conclusions of students are similar to conclusions already published in Aye et al., 2016c, which reads as "could be further improved". Therefore, the novelty of this paper could be questioned. Although the paper is well structured and concepts and exercises are described in detail so that the reader can get a good idea of the tool and the students' work sequence, I cannot recommend the publication unless substantial scientific findings are included in the paper.*

Even though methods applied are not new, RISKGIS can be considered as a new, simple and practical tool for students studying environmental risk beyond the traditional (paper-based or desktop GIS) approach, which is currently being used in the classroom. As we mentioned in Sect. 2 of the manuscript, due to the spatial nature of risk, it is often difficult to deal with realistic, geographically large and complex cases on paper in exercises. With RISKGIS, students can not only learn approaches used by experts but also gain insights in complexity of risk management through real case examples. Besides, through role-playing, students have an opportunity to experience different roles and perspectives of stakeholders in risk management. This is supported by four moments of

experiential learning theory (Kolb, 1984). Moreover, thanks to its advantage of being a web-GIS, it can be easily accessible from a web browser without needing to install additional GIS software. Notably, it can be freely adapted and reproduced as necessary due to its open-source modules and technologies, which is beneficial to the scientific, academic and open-source community in natural hazards. Some of its benefits are already highlighted by the reviewer, and yes, this tool could indeed be adapted for real decision makers in natural hazards and risk (as asked by the reviewer in the supplementary file). *Aye et al., 2016c* was the kind of tool meant for real decision makers in risk management, and in RISKGIS, we adapted it for students but with approaches used in Switzerland for relevant course exercises. As the reviewer questioned, we are also currently working on a web-GIS application for Canton Vaud authorities in Switzerland for risk management of natural hazards.

The findings of this research are strengthened and supported by feedback collected from students and observations in classroom. Probably in this current version, we did not formulate enough to highlight main contributions and findings (as also pointed out by another reviewer). We will revise it accordingly in respective sections. However, compared to *Aye et al., 2016c* and concerning the novelty of RISKGIS, in this research, the focus is placed on teaching and learning (as already explained and listed above). Thus, this study is different from *Aye et al., 2016c*. We are also using RISKGIS in this academic year, based on the feedback we obtained from students last year. Hence, there could again be improvements for the next year, considering it is an iterative and cyclical process in applying the platform with students.

- *As our comments indicate, the used terminology should be critically checked since it is not used consistently throughout the paper. Additionally, please have a native speaker do a detailed revision of the language. Text flow and comprehension need to be improved.*

Thanks for your kind suggestions and corrections. For the revised paper, we will make sure that the terms are consistent (such as "exercises" instead of "scenarios", which can be confusing as the term "scenarios" was also used to describe risk and alternative scenarios, for example, in exercises). We will also check the language so that the readability and flow of the paper is improved.

**Further comments:**

- *Especially for key vocabulary, choose one term and stick with it throughout. E.g. protection measures – why confuse the reader with "alternatives"? Or "scenarios" for "exercises" – simply use exercises in the whole paper to keep things clear.*

In RISKGIS, we use the term "alternative" as a combination of protection measures, i.e., to specify that there can be different protection measures in an alternative. We will revise accordingly so that the reader will not be confused.

- *Give a brief description of the Innovative Teaching Project.*

The Innovative Teaching Project (2016) is funded by the Innovative Teaching fund (Fonds d'innovation pédagogique) of the University of Lausanne, and the underlying question of this project is "How does an environmental risk system work?". The first objective is to make students understand that the evaluation of risks depends on the knowledge of the system to be evaluated and on the level of expertise, and to become familiar with approaches used by experts in natural hazards. This project concerns two courses: Environmental Risk, and Advanced Quantitative Risk and Vulnerability courses. The Environmental Risk course (9 sessions per semester) sets the goal to introduce environmental risks and their implications on the society from a quantitative viewpoint, while the Advanced Quantitative Risk and Vulnerability course (7 sessions per semester) is dedicated to the in-depth understanding of risk notions for a thorough risk calculation of natural hazards and it presents methods of Swiss Confederation among others including expert approaches such as impact-probability matrices. During the courses, a series of lectures are delivered accompanied with exercises as part of each session such as different types of hazards

and risk, risk calculation, statistics applied to natural phenomena in risk assessment and so on. Particularly in the Environmental Risk course, previously in exercises, interactive (spatial web) tools such as RISKGIS were not applied. Exercises that involve risk calculation, cost and profitability of mitigation solutions were carried out using synthetic cases and on paper. This is where a web-GIS solution like RISKGIS comes to play a significant role in classroom, allowing not only students to visualize, analyze and compute spatial data for risk estimation and management but also teachers to develop more concrete exercises using real case studies. As a part of this project, the web-GIS platform (RISKGIS) is therefore developed and applied progressively in courses replacing some of the paper-based exercises throughout the semesters (2016). Three teaching scenarios involving the application of RISKGIS are designed to support progressive and experiential learning of students. First, students calculate risk starting from hazardous situations such as earthquakes, landslides and debris flows. The calculations involves the vulnerability, elements-at-risk and their values, people, etc. Then, students work in group by sharing their thoughts and knowledge during the process, with an objective of taking decisions and establishing a strategy based on cost-benefit and multi-criteria analyses. The risk reduction is assessed and evaluated by groups and discussed within the course, allowing students to also have the possibility to give and receive feedback by his colleagues and teachers. Combing this (online and offline) participative approach with role-playing activities, students have an opportunity to experience like experts in real world and can learn from each other. This brings innovative pedagogical practices and values to traditional curricula and classroom setting. The findings through the empirical evaluation (i.e. feedback of students and observations in classroom) supported aspects of experiential learning, in which the learning experience of students are improved through group work, collaborative interaction, discussion and hands-on participation.

We will revise the respective section accordingly for a better explanation of the project: its objectives, teaching scenarios, approaches and added values.

- *What is the purpose of the questionnaires concerning the group work (e.g. figure 15)? What do you learn from the answers and how is this information useful?*

The purpose is to understand aspects related to group functioning and decision making, particularly in the third stage of the third exercise. Students played the roles of different stakeholders in decision making for selection of alternatives. The first questionnaire is for the first round of selection, in which each group of students (same role) makes the decision within group. This is to analyze how students with same role behave in a group and how working together in a group can benefit them in decision making. The answers were as we expected, supporting our findings in group functioning. The second questionnaire is for the second round of selection, in which all groups of students (mixed roles) come together and discuss to achieve a final solution. Here, we want to answer questions such as who influences the decision making process in a mixed group with different roles of stakeholders. The feedback results were consistent, and through this process, students were also able to grasp the real situation, conflicting interests of stakeholders and the complexity of a group decision making process in risk management. We will include this explanation and findings in the revised version.

- *In the case of the Brienz case study: Was the real solution presented and discussed in class? That would nicely round off the exercise, showing students what was decided by the experts and explaining why.*

Thanks for your comment. That would really be a good round off. Particularly in the third exercise, protection measures taken in the study area was briefly presented and explained. For the second exercise, as they still have to submit a group report within three weeks (if a report is good, we give a bonus note to students), we did not present the real solution in the beginning. For example, in this year, we presented an example case along the actual one so that they can have an idea.

- *Could the tool be adapted for real decision makers in natural hazard risk? Perhaps that could be a future aim for the project.*

Yes, it could be adapted. Contrary to the reviewer's comment before, *Aye et al., 2016c* is an example of such tool for real decision makers. However, in RISKGIS, we adapted *Aye et al., 2016c* for students but with approaches applied in Switzerland for relevant course exercises. We are also currently working on an online tool for Canton Vaud authorities in Switzerland for risk management of natural hazards, under the framework of another project.

**Technical corrections:**

- *Page 1, line 1: "...with a free and..."*
- *Page 1, line 9: "...is developed for students studying environmental risk..."*
- *Page 1, line 10: "...become familiar with..."*
- *Page 1, line 12: "...hands-on..."*
- *Page 1, line 12: "...To identify the potential and practicality of the ..."*
- *Page 1, line 14: "... semesters of the Environmental Risk and Advanced Risk and Vulnerability courses at the University..."*
- *Page 1, line 15-16: "...are conducted starting with the rapid risk ... exercise and moving on to the more complex risk ... exercise incorporating different case studies..."*
- *Page 1, line 17: "...are asked to take a test, complete feedback questionnaires or write group reports on the Moodle platform in order to evaluate the exercises, the RISKGIS platform and the performance of the students..."*
- *Page 1, line 20: "...of 64/100 are achieved..."*
- *Page 1, line 21: "...and feedback from the students..."*
- *Page 1, line 26: "...van Westen, 2013). Rapidly developing technologies such as GIS play an..."*
- *Page 2, line 2: "...achieving goals of science education such as utilisation of technology and development of..."*
- *Page 2, line 4: "...the evolution of the web and with the advancement of technology it has..."*
- *Page 2, line 10: Either "in teaching and learning" or "in instructional settings"*
- *Page 2, line 11: Accessibility to hardware: you still need a computer to access a webGIS, so how is the need for hardware reduced?*
- *Page 2, line 12-13: "...platforms can be easily accessed from..." or "...platforms are easily accessible from web browsers without purchasing GIS software. ..."*
- *Page 2, line 12: "limited resources of the Lab": What resources, what lab? See suggestion above (p. 2, lines 12-13)*
- *Page 2, line 19-20: "interactive, active, activity" in the same sentence – try to eliminate at least one. E.g. "task" instead of activity*
- *Page 3, line 14: ..and in the following paper: Consider using "exercise" instead of "scenario" since scenario has a different meaning in the risk context.*
- *Page 6: line 20: Explain what you mean with "Jigsaw".*
- *Page 7: line 5-6: eliminate parentheses: either use the word or don't (exception: IRM).*
- *Page 8, figure 2 and rows 3-5: Choose more unmistakable terminology for the figure (and in general). E.g. instead of "alternative formulation" use "planning of measures" or "choosing a course of action". Also, make sure your explanation of the figure (rows 3-5) uses the same words as appear in the diagram. Add numbers to the figure so it is very clear which step is which and the reader doesn't have to do the matching herself."*
- *Page 8, line 10: What is meant by hazard layer? I would rather suggest the term "scenario".*
- *Page 9, line 8: The value 5000000 CHF needs to be further explained. I recommend to replace it with a factor since 5000000 CHF is a value used in Switzerland. The VSL approach reflects the societal willingness to pay for averting a fatality, which is closely related to a specific country.*
- *Page 9, line 11: 12 hours in each of 365 days per year.*

- *Page 9, line 15-21: The unit for individual risk should be 1/year in my eyes. It translates as how often per year, a particular person (a person living in object i) is likely to die. With the unit "deaths/year" you are suggesting that the individual risk says how many people die per year, which is what the collective risk (non-monetised) is about. By using the unit 1/year, you can eliminate the irritating "1" in equation 3.*
- *Page 9, line 28-30: year in citation UNISDR is missing.*
- *Page 10-11, line 26-29: Use a variable (e.g. CBR) instead of an actual ratio ($N/K_{tot}$) to name the cost-benefit ratio in equation 5. $K(j)$ is confusing, especially since up until now, $j$ has been the index for a particular hazard. Use $K_{tot}$ throughout. Even if it is fairly obvious, explain R(before) and R(after) when you list your variables below equation 5.*
- *Page 11, line 14 and following: Alternative and criteria are not capitalised in this context.*
- *Page 11, line 23: Ideal not idea*
- *Page 11, line 20: Explain $L_p(x_p*)$ in equation 7.*
- *Page 12, line 10-13: Why are some words italic? Also the case in following lines.*
- *Page 12, line 23: What is the benefit of a remote area?*
- *Page 12, line 26: Wouldn't that be a return period of 500 years, then? 25 years may not make a big difference on that scale but it is confusing for the reader.*
- *Page 12, line 27: The term "location site effects" does not make sense.*
- *Page 14, figure 4: There should not be any red bars (even if they are very slight) for a 0 value. Use the word points (or an equivalent) instead of notes (in the text above and below the figure as well as in the figure itself).*
- *Page 14, line 6: As it turns out in line 11 and following, you are not asking questions but giving statements which students can agree or disagree with. Hence, please change the word "question" as well as the abbreviation "Q" for the statements (applies to the rest of the paper, too). Mention the Likert scale here as well, not only later on in the paper. It may give the impression, that you are using two different approaches.*
- *Page 14, line 9: You might mention that a SUS score above 68 is considered above average…*
- *Page 15, line 9-16: Consider putting these pros and cons into a bullet point list for a better overview. Consider it, too, for the presentation of the other results in the following paper.*
- *Page 26, line 12: This question does not fit well with the scale "not at all to absolutely". Consider rephrasing it or adding a second scale.*
- *Page 27, line 11: Explain your scale of 1 to 5 in words in the text, not just the figure caption. Again, not all the questions are suitable for "not at all to absolutely".*
- *Page 27, figure 15: Try to avoid 3D graphs.*
- *Page 28, line 23: Either use the word collaborative or don't but don't put it in parentheses – this makes for cumbersome reading.*
- *Page 28-29: A detailed repetition of the questionnaire results is not appropriate in the discussion. Generalise and mention only the key points. Also, the suggestion of the exercises running parallel to the course should not appear for the first time in the discussion – this belongs with the questionnaire results and can be briefly mentioned here as a potential future development for the course..*

Many thanks again for your corrections and suggestions. We will correct them as suggested in the revised paper.

- *Page 12, line 16: Why did so few students comply? Could it be that only a certain "type" of student took the quiz and answered the questionnaire, thus distorting your results?*

Because it was not obligatory for students to answer the quiz and feedback questionnaire in the first exercise. For example, during the third exercise, as there were only 13 Master students, the whole exercise was done on the same day and we asked students to fill questionnaires (in paper format) at the end immediately.

- *Page 20, line 19: Was the consideration of ecological aspects already part of this exercise?*

Yes, in the last stage of the third exercise. Different aspects (social, economic and environmental) are considered in the selection of alternatives. In this stage, students played different roles of stakeholders (in groups) and ranked alternatives based on their preferences on decision criteria.

- *Page 26, line 20: What is moderate, what is high severity?*

58% of students had conflicts in prioritizing decision criteria within the group. However, when it comes to the severity level of conflicts, on a scale of 1 to 5, only 38% of students gave a score of 3 (moderate) while no students indicated a score of 5 (high). We will rephrase it more clearly in the revised version.

- *Page 28, line 8: "...cost was the better choice despite the limitation of risk for human beings." I don't understand this argument.*

It was because cost was an important criteria for students who played the role of mayor, while the technician group considered it as the least important criteria in the selection of alternatives. This comment was given by students who played the role of geologists. Their alternative was ranked second marginally due to its cost, which was much higher than the winning alternative, while their alternative provided a better safety to affected people and buildings in the area. During the final round of selection, only if the committee led by the mayor agreed to weigh less on cost criteria, the alternative of geologists would have been selected. This means students clearly understood that there are trade-offs in selection, and that it is important to share a common goal among all involved stakeholders. We will include it in the revised version.

---

## Author Comment (AC2) · 4 May 2017

**General/Specific comments:**

- *the manuscript 'Learning risk management of geohazards in practice with free and open-source web-GIS based platform: RISKGIS' describes and provides an overview of an teaching project with students at bachelor and master level. The web-GIS based platform aims that students learn and understand environmental systems with a focus on geohazards and risk as well as to get familiar with different expert approaches applied in Switzerland. The platform should also allow the lecturer to evaluate the performance of the students and the web-tool. The platform is tested and illustrated by three different examples applied in courses of the University of Lausanne presenting different settings such as bachelor and master level, individual task for students or group work, and high and low number of students. The main results of this study are the evaluation by the students of the different courses using the platform. The idea of the web-GIS platform in teaching and preparing students for their applied work after university has a high potential and is attractive for students by using the approach 'learning by doing'. However, the presentation of this study in the manuscript is a detailed description of the teaching projects, yet it lacks of new scientific insights in the field of education, web-GIS based platforms or risk analysis considering also the existing publication of this project (Aye et al 2016c).*

Thanks for your summary and comment on the potential of the RISKGIS platform. As we already explained to the reviewer 1 (see specific comments), we would like to mention that this paper is not an additional paper of *Aye et al., 2016c,* and they are not carried out under the same project. *Aye et al 2016c* was neither designed for students nor for respective courses at the university. Instead, it only served as one of the motivations for the development of this teaching project, since we tested *Aye et al 2016c* with students for the evaluation of the prototype platform (*Aye et al 2016a and Aye et al 2016b)* which was meant for experts and decision makers in three European case studies. As we mentioned in Sect. 2, students expressed their interests in performing exercises using such kind of interactive tools. As a result, also considering pedagogical needs, the RISKGIS platform was evolved through the adaptation of our existing works. We will revise this clearly in the manuscript so that the reader would not be confused.

For clarification, new features of RISKGIS are also listed below, compared to the prototype of *Aye et al., 2016c*. In this study, RISKGIS is specifically designed for:

- certain courses and practical exercises in university for education;
- risk concepts and methods applied in Switzerland;
- rapid risk estimation based on qualitative (vector) hazard intensity map and OpenStreetMap data;
- cost estimation of measures;
- manually edition of new input maps for risk estimation (such as hazard intensity and buildings map);
- cost-benefit calculation of different risk mitigation scenarios and
- additional features such as registration, customized data sharing and interfaces for students and teachers.

Besides, pedagogical scenarios for progressive learning are designed and implemented in this study, starting with the rapid risk calculation and moving on to the more complex risk management concepts incorporating real events of natural disasters as case studies. In addition, test quiz, group assignments and various questionnaires are integrated via the Moodle platform for the purpose of evaluation.

We agree that the presentation of the manuscript should be improved so that contributions and findings of this study are shed to light. Nevertheless, we believe that this study is worth presenting to the scientific and academic community, considering also that it can be reproduced due to its open-source modules and technologies. As the first reviewer questioned, it can be adapted for real decision makers in risk management (*Aye et al 2016c* is such an example). In the final publication, a link to the source code of the present application can be made available. Even though existing methods are applied in this study, RISKGIS itself can be considered as a new tool beyond the traditional (paper-based or desktop GIS) approach which is currently being used in classroom, and with a bit of web-GIS training, students can benefit from using this platform. Besides, as highlighted in Sect. 1, most web-GIS

tools in education are used for the visualization, dissemination and mapping of spatial data with little capacity for interactive analysis and computing, particularly in this field. Through the presented pedagogical framework applying RISKGIS and OpenStreetMap data, students can not only become familiar with approaches used by experts but also perform a rapid risk calculation of possible different scenarios with available data. Since, in reality, availability of quantitative hazard intensity maps and complete inventory of elements-at-risk are rather limited to carry out a full quantitative risk analysis. Moreover, teachers can also monitor and trace students' performances and progresses in RISKGIS as all data and results produced by students are centralized in the system.

- *For example section 2 highlights some pedagogical concepts and aims, however the questions of the courses evaluation answered by students is not clearly related to these concepts and seems to be strongly based on the standard lecture/exercise evaluation at universities. I miss a clear and structured approach to evaluated a) pedagogical concept/aim of the project and b) the usability of the web-platform. I cannot recommend the current version of the manuscript for publication due to the missing added value for the scientific community as described above. Furthermore, I would recommend re-structuring the manuscript according to focused research questions, considering a clear structure for methods, results and discussion (e.g. questions of the evaluation are part of the methods section) and re-writing a critical discussion section according to the new research questions and highlighting the pro and cons of the web-GIS platform in education. Moreover, I suggest language editing and check used technical terms according the consistency within the manuscript. Some further comments are highlighted in the attached file.*

In section 2, we described pedagogical needs, scenarios for progressive learning, advantages of the web-GIS solution, and pedagogical added values of the project (see also the brief summary of this Innovative Teaching project in page 2-3 of our response to the first reviewer). The aim of the project is to support the implementation of various pedagogical scenarios under the framework of the Environmental Risk and Advanced Quantitative Risk and Vulnerability courses of the university. For this purpose, RISKGIS is developed by adapting the existing work of authors. Some of the paper-based exercises are replaced with RISKGIS, and real case studies of hazard events and approaches used by experts are used in RISKGIS to support the experiential learning of students. This is further complemented by activities such as test quiz, discussion, group work, role-playing and hands-on participation, allowing students to develop skills in problem solving and critical thinking. The findings of this study are then supported by empirical evaluation, based on the observations and feedback collected from students. The usability of the web platform is evaluated by using the SUS method (i.e., a ten-item questionnaire with a simple 5-point scale from Strongly agree to Strongly disagree), as presented in the manuscript.

Regarding the evaluation questions, in the exercise feedback, for example, we asked questions not only about the impact of exercises but also what they have learnt from doing such exercises. Since, most of the times, students only want to finish an exercise without actually trying to understand the concepts behind. Feedback results showed that exercises encouraged students to analyze a problem at hand and reflect on different aspects of the presented topic, for example, in proposing potential protection measures and why these measures should be implemented compared to others. This supported the learning strategies we considered in developing the experiential learning (see Sect. 2). Similarly, the findings of group functioning questionnaires showed that students benefit from each other (despite there were some conflicts) by working together in a group and by playing the roles of stakeholders in risk management. In addition, learning through a web-GIS, students can benefit from learning some aspects of the GIS as there are undergraduate students who never got to learn GIS before using RISKGIS in exercises. This made possible to introduce web-based tools and approaches in classroom and exercises, and students only need to bring their own devices without needing to install additional software for computing.

As we also replied to the first reviewer, probably we did not formulate clearly in Sect. 2 and the following result and discussion sections of three exercises. In Sect 3, we mainly presented background methods developed in RISKGIS rather than evaluation methods for performance of students, exercises and the platform. Instead, evaluation questionnaires were presented separately under respective exercises along with the feedback of students. We plan to

revise respective sections accordingly to highlight pedagogical achievements and main findings in a clear and structured manner, supported by feedback of students and observations in classroom. As suggested, we will also check the language and consistency of technical terms in the revised paper.

**Further comments:**

- *Page 10, line 10: Is this the same reference as EconoMe 2015?*

EconoMe (2015) is the reference to formulas, and OFEV (2016) referred to the actual website of the EconoMe application.

- *Page 11, line 4: see also Fuchs /Mc Alpine.*

Thanks for providing the reference.

- *Page 12, line 12-14: provide this information.*

Later in Sect. 4.1.3 (Results and discussion), we have included this information (quiz and feedback questionnaire) along with feedback results of students. In the revised version, we will provide this information under the relevant section, separating it from the results and discussion.

- *Page 12, line 28: who did this? The students in the exercise or was this prepared by the instructors?*
- *Page 12, line 30: by whom?*
- *Page 16: line 25: by whom?*

The input data were prepared by the instructors including the extraction of buildings from OpenStreetMap.

- *Page 14, line 5-11: this should be part of the methods expect the results.*
- *Page 20, line 2: again the questionnaire is part of the methods not the results.*
- *Page 25, line 2: some remark - parts should got to the method section.*

Thanks for your suggestion. As we mentioned before, we will separate the evaluation questionnaires from the result and discussion sections of exercises.

- *Page 17, line 6: the list does not include questions.*

Thanks for your comment. We will correct it accordingly.

- *Page 18, line 10: this paragraph is a repetition to already described procedure.*

This is because we had a separate sub-section for the stages of the exercise, illustrating how the students performed the exercise in step-by-step using the RISKGIS platform. These stages followed the main procedure we presented in Sect. 3 and Figure 2, except the last Multi-Criteria Analysis component. We will revise it accordingly.

- *Page 19, Figure 8: you use the colour the Swiss hazard map but the information should be the intensity of certain scenario. I think you should use different colours otherwise the student will think that intensity maps is the same than a hazard map (matrix frequency/intensity). You can not derive risk analysis form the Swiss hazard maps.*

Yes, colors represent three intensity classes of a hazard intensity map. Indeed, using the same color styles as the Swiss hazard map can be confusing. We will use different colors in next exercises with students and will also update the figure accordingly in the manuscript.

- *Page 20, line 7: of what?*

We wanted to know if the exercise was useful for students in learning and understanding of the presented topic (contents of the exercise).

- *Page 21, line 5: would this not be an essential aspect for teaching? it seems that the tool is for several parts a black-box.*

We agree that this is essential for students to be able to answer questions such as which measures to select, what are their advantages and disadvantages, and how these measures could reduce the hazard or risk of a given area? The tool provides assistance to students in performing a rapid risk calculation before and after protection measures. However, it does not necessarily model a new hazard map or a new risk situation automatically after protection measures. This is because of the level of complexity involved, and more detailed data and information of the territory are also required, varying according to different hazard types in exercises. Alternatively, other existing tools and simulation models can be jointly applied. Otherwise, additional information and materials are needed to provide during the course and exercises. For example, in this academic year, we provided indicative costs for cost estimations of measures, along with examples of possible measures which can be used for protection against debris flows.